# Towards Improving the Outcomes of Multiple Ovulation and Embryo Transfer in Sheep, with Particular Focus on Donor Superovulation

**DOI:** 10.3390/vetsci9030117

**Published:** 2022-03-04

**Authors:** Sami Ullah Khan, Muhammad Ameen Jamal, Yanhua Su, Hong-Jiang Wei, Yubo Qing, Wenmin Cheng

**Affiliations:** 1Faculty of Animal Science and Technology, Yunnan Agricultural University, Kunming 650201, China; samiullahakbar4@gmail.com (S.U.K.); drameen007@gmail.com (M.A.J.); 2Yunnan Key Laboratory of Porcine Gene Editing and Xenotransplantation, Yunnan Agricultural University, Kunming 650201, China; 2013003@ynau.edu.cn (Y.S.); hongjiangwei@126.com (H.-J.W.); 3College of Veterinary Medicine, Yunnan Agricultural University, Kunming 650201, China; 4Xenotransplantation Research Engineering Center in Yunnan Province, Yunnan Agricultural University, Kunming 650201, China

**Keywords:** hormonal protocols, intrinsic and extrinsic factors, embryo recovery and transfer, ovine

## Abstract

Considerable improvements in sheep multiple ovulation and embryo transfer (MOET)protocols have been made; however, unlike for cattle, MOET is poorly developed in sheep, and thus has not been broadly applicable as a routine procedure. The tightly folded nature of the ewe cervix, the inconsistent ovarian response to various superovulatory treatments, and the requirement of labor to handle animals, particularly during large-scale production, has limited the implementation of successful MOET in sheep. Moreover, several extrinsic factors (e.g., sources, the purity of gonadotrophins and their administration) and intrinsic factors (e.g., breed, age, nutrition, reproductive status) severely limit the practicability of MOET in sheep and other domestic animals. In this review, we summarize the effects of different superovulatory protocols, and their respective ovarian responses, in terms of ovulation rate, and embryo recovery and transfer. Furthermore, various strategies, such as inhibin immunization, conventional superovulation protocols, and melatonin implants for improving the ovarian response, are discussed in detail. Other reproductive techniques and their relative advantages and disadvantages, such as artificial insemination (AI), and donor embryo recovery and transfer to the recipient through different procedures, which must be taken into consideration for achieving satisfactory results during any MOET program in sheep, are also summarized in this article.

## 1. Introduction

Sheep are the most efficient of all domestic species. Sheep possesses various unique characteristics, such as the ability to convert the natural forage of extreme habitats into valuable protein [1], and the adaptability to harsh environmental conditions, enabling survival on sparse desert ranges with low water availability [2]. Additionally, sheep can contribute directly and/or indirectly to various household and economic securities, such as food, socio-cultural wealth, clothing, and crop by-products [3]. Improvement in soil fertility and locally grown vegetation are other advantages of sheep farming [4]. The FAO statistical database (FAOSTAT, 2020; http://www.fao.org/faostat/en/#home; accessed on 15 March 2021) reports 1239 million sheep heads in the global population in 2020, which increased by approximately 15% in the last 15 years [5]. However, this estimated population size has remained almost stable over the last three decades (FAOSTAT, 2020). Assisted reproductive technology (ART) plays a central role in allowing rapid genetic progress, providing flocks with the potential of producing milk round the year, shortening reproductive cycles, and increasing fertility and prolificacy according to production purposes. Therefore, it is important to properly develop and implement various reproductive biotechnologies that will enable this industry to be more profitable for producers [6].

Multiple ovulation and embryo transfer technology (MOET) is a reproductive technique that primarily aims to fertilize several oocytes in a shorter time period to produce more viable embryos, which are transferred into the recipient, resulting in a higher birth rate [6,7]. This technology is based on estrus synchronization and hormonal stimulation of donor animals, followed by AI, and, finally, the collection and transfer of viable embryos to the recipient animal [8,9]. MOET is playing a central role in the global trade of genetic resources, conserving endangered species, maintaining elite dairy herds, minimizing the risk of exotic diseases and the cost of production, and eliminating transportation stress [10,11]. Hence, considering these advantages, biotechnologists aim to conduct a great quantity of scientific work to improve the efficiency of this technology in livestock production [11]. However, this technology is still not completely applicable in sheep due to their relatively small physical size and complexity regarding the anatomy of the tightly folded cervix, which inhibits rectal manipulation [12,13]. The most challenging aspects of this technology is the inconsistent ovarian responses of donor sheep to various hormonal treatments, difficulty in embryo recovery, and the reproductive status of the recipient when transferring viable embryos [14,15,16].

Superovulation is an important step in any MOET program, in which the maximum oocyte number can be produced by the same animal through injecting various gonadotropin hormones. During superovulation, intravaginal sponges are inserted for 12 to 14 days, followed by various exogenous gonadotropins, usually starting two days before the withdrawal of vaginal pessary [7,17]. This stimulates follicular growth, which ultimately leads to multiple ovulations. However, the ovulation rate varies according to various hormonal protocols, dose of gonadotropins, number of injections, and protocol rationale. Irrespective of this, donor breed and age, season and location, nutritional status, and donor health status also affect the ovulatory response [18,19]. Therefore, it is generally acknowledged that there is a lack of well-established superovulation protocols in sheep. Other factors affecting superovulation in sheep include the failure, or extremely low fertilization rates, of AI, and low embryo recovery [10]. This review discusses the outcomes of commonly used hormones in different superovulation protocols and their applications, and outlines the current understanding of the factors affecting the superovulatory response in ewes. We further elaborate upon the influence of AI, and embryo recovery and transfer into recipient ewes, which is more practicable in cattle than sheep.

## 2. Animal Management and Selection

MOET efficiency generally depends upon the reproductive status of donor and recipient sheep, which relies on their housing and management [20]. The experimental animals should be clinically healthy and have free access to water and salt adlibitum [21,22,23]. Both donor and recipient sheep should be provided with a separate housing shelter, as excess stress can affect the superovulation response. The animals should be vaccinated against any infectious diseases, and their health status should be properly maintained [10].

Furthermore, the control of body condition is obtained through proper nutritional management, as nutritionally induced metabolic status affects reproductive parameters [20,24]. Diet supplements containing proteins and energy enhances the potential for viable embryos in donor females [25], whereas under nutrition during the period of superovulation reduces the number of viable embryos [24]. However, the influence of nutrition is more important in the recipient in terms of pregnancy, as it impacts the growth and development of the conceptus [20].

Another major aspect is the selection of donors and recipients for superovulation. Ewes selected must be between 3–6 years of age, because at this time they have undergone sufficient physical development, and ovary function is optimal [11,15]. The body condition score (BCS) is critical for determining the ovarian response of the donor and embryo survival in the recipient after implantation [26]. To optimize the ovarian response, a BCS between 3.0 and 3.5 (BCS, 1–5 scale, 1 = emaciated, 5 = obese, [27]) is required [12,23,28]. The food and water conditions during embryo collection and transfer are described in the section “Donor embryo recovery and transfer to recipient”.

## 3. Hormones for Superovulation

Superovulation is typically achieved using gonadotropic hormone preparations that promote the development of subordinate follicles in order to ovulate, or through inhibin immunization to eliminate the inhibitory mechanism of the dominant follicle. Various gonadotropic preparations (FSH and eCG) are widely applied to induce superovulation in sheep [29,30]. Other hormones, such as horse anterior pituitary (HAP) extracts, human menopausal gonadotropin (hMG), gonadotropin-releasing hormone (GnRH), and human chorionic gonadotropin (hCG) have also been applied, although less frequently, during sheep superovulation [31,32]. The superovulation responses associated with each hormonal stimulation are detailed below.

### 3.1. FSH and Superovulatory Response

FSH is the most commonly used hormone for inducing superovulation in sheep and other livestock [29,33]. Generally, it is administered via multiple injections (6–10) at 12 h intervals over a period of 3–5 days during the follicular phase [12,15,34]. Variation in ovarian response to this procedure has been observed in sheep (Table 1). Collectively, maximum rates of ovulation (15.9) and embryo recovery (10.7) were observed using the conventional FSH protocol [35]. The major disadvantage of FSH is its short half-life, necessitating daily administration [36,37,38]. In addition, multiple injections are time consuming and stressful for animals, which may be detrimental to the female reproductive performance [35]. Furthermore, the potential for premature luteal regression, substantial individual variability in ovarian responses, and the low quality of recovered embryos, are also limiting factors of FSH [33,39].

To cope with these shortcomings, various researchers have attempted to replace the classical FSH multiple injection protocol with either a single FSH injection [40,41,42,43] or a single injection of FSH in combination with eCG [29,35,44,45,46,47]. However, no significant difference between single and multiple injections has been found for ovulation rate (10.2 vs. 10.8), embryonic recovery (9.9 vs. 10.5), or fertilized embryos (5.2 vs. 3.9); however, the proportion of good-quality embryos (5.1 vs. 2.9) was reported to be comparatively higher in the simplified than the conventional superovulation protocol (Table 1) [36].

Furthermore, a comparable ovulation rate (14.5 vs. 15.9, *p* > 0.05) and embryonic recovery (11.3 vs. 10.7, *p* > 0.05) was obtained for the FSH plus eCG single injection and FSH multiple doses, respectively [35]. FSH plus eCG in a single injection not only resulted in a greater proportion of viable embryos, but also the advanced onset of estrus and LH peak [29]. A few successful simplified attempts have also been made in cattle [48,49,50,51,52] and goats [8].

These findings indicate that FSH in combination with eCG in a single injection could produce an acceptable ovarian response. However, due to the lack of studies on this simplified superovulation protocol in sheep, the endocrinological basis of this protocol remains poorly understood.

### 3.2. eCG and Superovulation Response

Equine chorionic gonadotropin (eCG) is extensively used for superstimulation in sheep. This hormone is typically injected via a single dose for 1–2 days before estrus synchronization [53]. Although eCG is cost-effective and can easily be applied to an open flock with minimal bodily stress [54], dose-dependent responses may lead to an increased number of persistent large follicles [54] and premature corpus luteum regression [55]. Furthermore, it also influences the pattern of steroidal hormone synthesis, thus disrupting sperm and gamete transport, as well as preimplantation embryo development [14]. To potentiate the ovarian response, the combination of eCG and FSH has been applied, and increased ovulation rate (14.2 vs. 6.2; *p* < 0.05) and embryonic recovery (5.2 vs. 1.0; *p* < 0.05) was observed [53]. A significantly higher ovulation rate (13.8 vs. 6.2) and embryo recovery (8.4 vs. 1.0) has been reported for FSH/eCG combined administration compared to FSH dissolved in saline divided into four equal doses [29]. No significant difference in the mean number of corpora lutea (CL; 8.7 vs. 9.4), transferable rate (85.3% vs. 88.8%),or degenerated embryos (5.4% vs. 4.8%) was observed; however, the average number of transferable embryos (5.5 and 6.6; *p* < 0.001) was significantly higher in the eCG plus FSH group than with eCG alone [56]. Similarly, eCG at a dose rate of 800 IU with 12 or 16 mg FSH (multiple injections) obtained a higher ovarian response than with eCG alone (14.8 vs. 19.1 vs. 3.5; *p* < 0.05) [57]. These results demonstrate that the administration of eCG combined with FSH in a simplified form can provide a high superovulatory response as compared to the single injection of eCG, injected 1–2 days before sponge removal.

### 3.3. Other Rarely Used Hormones

Some investigators also tested alternative gonadotropin preparations for superovulation in sheep, such as HAP, hMG, GnRH, and hCG. For example, HAP, obtained from the pituitaries of slaughter material, has been reported to induce a comparable response to p-FSH conventional protocols in sheep [58,59,60]. This hormone reduced the excess of large, non-ovulatory follicles, and also improved the quality of transferrable embryos [14].

In addition, hMG has been shown to produce a comparable ovulation rate (10 vs. 9.9; *p* > 0.05), rate of transferable embryos (84% vs. 80%), and fertilization rate (86% vs. 95%; (*p* > 0.05) to that of FSH [31]. Similar comparable responses of hMG with FSH have also been obtained [61], and in one case, hMG produced higher ovulation rates and improved embryo quality than obtained through FSH [14]. However, this hormone has not been extensively utilized for superstimulation due to its high cost [62].

Several researchers injected GnRH after sponge removal in order to improve the synchronization of the ovulation of follicles when injecting FSH or eCG. It has been reported that the association of longer progesterone exposure with GnRH administration is an alternative method to improve oocyte fertilization rates, particularly during fixed-time insemination [9]. Similarly, GnRH administration 24h after sponge withdrawal increased ewes ovulation rate when treated with eCG. The number of recovered embryos from ewes treated with eCG plus GnRH (4.3 vs. 1.06; *p* < 0.05) was higher than in ewes treated with eCG alone [63]. Similarly, in Santa Inês ewes, donors were synchronized with an insertion of a progesterone controlled-internal drug release (CIDR) device for 14 days, exchanging with a new CIDR on day 7, and administered PGF_2α_. On day 12, the superovulatory treatment was initiated using 133 mg of pFSH (Folltropin) at eight decreasing doses twice daily. On day 14, at the time of CIDR removal, ewes were divided into the following three groups: control group (progesterone device withdrawn at day 14); 12h P4 group (progesterone device maintained for an additional 12 h, i.e., until day 14.5); 12h P4 GnRH group (progesterone maintained for an additional 12 h, i.e., until day 14.5, plus a GnRH agonist). The results showed that a higher fertilization rate (77% vs. 34% vs. 41%; *p* > 0.05) was found in the group of ewes with progesterone devices maintained for 14.5 days plus a GnRH agonist during the last FSH injection, than the other two groups [9]. In addition, GnRH administration at the end of superovulation using FSH resulted in greater synchronization between the first and last ovulation events [9]. Similarly, at the end of the superovulation protocol, the administration of GnRH increased the fertilization rate, which ultimately enhanced the proportion of viable embryos [64].

A short-term protocol has been described, consisting of intravaginal sponges inserted for six days in Santa Inês ewes. On day 5, 300 IU of eCG and 37.5 μgd-cloprostenol were injected. The GnRH agonist at 24 h showed no benefits; however, using a GnRH agonist at 36 h more efficiently synchronized ovulation and promoted the desired environment, with the absence of dominant follicles after ovulation. Therefore, combining a GnRH agonist at 36 h after sponge removal, as in the short-term protocol, with the start of superovulatory treatment at 80 h, may be recommended in ewes. The GnRH agonist at 24 h compared to 36 h after sponge removal showed no estrus response (0.0% vs. 78.0%) [65]. The GnRH at 36 h after sponge removal efficiently synchronizes ovulation in ewes treated with FSH due to the improved synchrony in ovulation, promoting the absence of the dominant follicles, which improves fertilization rate and increases the production of viable embryos [64].

hCG is structurally similar to LH, and has been shown to increase total CL weight in ewes [66]. One explanation for this could be that hCG administration during the early luteal stage encouraged supplementary CL formation [67]. Moreover, administering hCG and vaginal sponges has been shown to have positive effects on lambing rates [32].

**Table 1 vetsci-09-00117-t001:** Comparative ovarian responses of simplified and traditional superovulation protocols.

Breed	Superovulation Protocol	OR ^1^	ER ^2^	FR ^3^ (%)	TRR ^4^	References
Awassi breed	FSH decreasing doses	8.75 ± 0.4 ^ac^	4.83 ±0.6	-	-	[68]
	eCG 1200 IU, single dose	5.66 ± 0.4 ^bd^	4.66 ± 0.6 ^a^	-	-	
Corriedale and Bond	In simplified protocol, FSH 180 mg	10.2 ± 3.4	9.9 ± 3.6	52.5	5.1 ± 4.9	[36]
	6 FSH administered twice daily	10.8 ± 4.7	10.5 ± 5.2	37.1	2.9 ± 2.9	
Corriedale	oFSH + eCG, single injection	13.8 ± 1.9 ^a^	8.4 ± 1.4 ^a^	64.2 ^b^	-	[29]
	oFSH dissolved in saline, divided into 4 equal doses	6.2 ± 1.1 ^b^	3.1 ± 1.1 ^b^	45.9 ^b^	-	
	oFSH dissolved in 30% polyvinylpyrrolidone, single dose	4.7 ± 1.0 ^b^	3.2 ± 1.1 ^b^	89.7 ^a^	-	
	oFSH 72 h before and 12 h after sponge removal, 8 decreasing doses	10.7 ± 0.9 ^a^	5.5 ± 0.8	93.9	-	
Fine wool Merino	FSH,7 decreasing doses 48 h before sponge removal during breeding and non-breeding season	13.9 ±0.8 ^a^ 11.3 ±1.8 ^a^	6.0 ± 0.5 ^a^ 3.5 ± 1.0 ^b^	-	-	[69]
	FSH 70 mg + eCG, single dose 48 h before sponge removal during breeding and non-breeding season	3.2 ± 1.2 ^b^ 6.0 ± 1.1 ^b^	1.2 ± 0.6 ^b^ 1.6 ± 0.5 ^b^	-	-	
Merino breed	eCG + 11.5 mg pFSH, 6 decreasing doses	14.2±1.2 ^a^	5.2± 1.9 ^a^	58.3 ^a^	-	[53]
	eCG, 1200 IU	6.2 ± 0.8 ^b^	1.0 ± 0.5 ^b^	26.3	-	
	eCG, 1600 IU	11.0 ± 3.0 ^ab^	1.2± 0.6 ^b^	19.2	-	
Ojalada	280 IU pFSH, 6 decreasing doses	15.9 ± 2.0 ^a^	10.7± 1.7 ^a^	86 ^a^	-	[35]
	210IU pFSH + 500 IU eCG, single dose	14.5 ± 2.1 ^a^	11.3 ± 1.8	76 ^a^	-	
Sarda	250 IU pFSH, 4 decreasing doses	11.8 ± 4.0 ^a^	8.80 ^c^	81.7	-	[19]
	125 IU pFSH + 600 IU eCG	8.05 ± 3.8 ^b^	4.82 ^d^	82		
Suffolk	eCG, 750–1000 IU, single injection	7.7 ± 1.4	3.5 ± 1.6	-	-	[70]
	FSH, 20–24 mg, multiple injections	8.4 ± 0.4	5.3 ± 0.5	-	-	
Xinji fine wool	FSH at the rate of 60, 50, and 30 IU per injection on days 1, 2, and 3	9.67 ± 1.93	7.85 ± 2.4	-	4.52 ± 2.5 ^b^	[71]
	150 mg Folltropin-V, twice daily at 35, 25, and 15 mg per injection on days 1, 2, and 3	12.47 ± 1.5	9.27 ± 1.8	-	7.86 ± 1.75 ^a^	

^1^ OR, ovulation rate; ^2^ ER, embryos recovered; ^3^ FR(%), fertilization rate; ^4^ TRR, transferrable embryos. Values with different superscripts within the column show a significant difference at *p* < 0.05.

## 4. Factors Affecting Outcomes of Superovulation

Various factors severely limit the practicability of MOET. Therefore, the manipulation of these factors may be key to attaining an improved ovarian response. The major factors are discussed in this section.

### 4.1. Ovarian Follicular Dynamics and Estrus Synchronization

The primary factor responsible for the variability in ovarian responses to any superovulation treatment is the ovarian status (the number and size of the antral follicles and CL on the ovaries) at the onset of an exogenous hormonal regimen. Furthermore, synchronization with follicular wave emergence has a major impact on the ovarian response to superovulatory protocols in sheep [72,73]. In sheep, a follicular wave is defined as the emergence or growth of 1–3 follicles from a pool of small follicles (2–3 mm in diameter) on the ovary, reaching >5 mm in diameter before regression (anovulatory wave) or ovulation (ovulatory wave). Usually, three follicular wave patterns are most prominent during the 17 day estrus cycle in sheep [74,75], and each follicular wave emerges every 4–5 days during the breeding season as well as during seasonal anestrus. The emergence of each follicular wave is associated with a transient peak in serum FSH concentration, which appears to be essential for follicle wave emergence, and each peak lasts 3–4 days [74,76]. Hence, the physiological status of donor ovaries at the onset of superstimulation plays a central role in determining the ovarian responses to exogenous hormones. Therefore, the synchronization of estrus is an important tool that can enhance the efficiency of ARTs. The traditional protocols for the synchronization in sheep are based on progestogen/progesterone treatment in the form of intravaginal implants, such as sponges and CIDRs [17,77]. Intravaginal sponges are usually inserted for 11–14 days, and ewes usually show estrous signs 24 to 48 h after sponge removal (Table 1). Through intravaginal sponge application, the estrus response and fertility varies greatly depending on multiple factors, such as management and mating system [78].

Although the synchronization of estrous facilitates MOET and optimizes its results, the unpredictability of the moment at which follicular waves emerge is the main limiting factor in predicting the superovulation response. A study was conducted to synchronize ovulation in ewes by cloprostenol injection, in order to evaluate follicle number per day and the superovulation response. The average follicle number on the ovaries between days 0 and 4 of the estrous cycle was measured daily viaultrasonography [79]. Ewes with higher follicle numbers (≥8) at the beginning of the estrous cycle had more CL and high-quality embryos than ewes with lower follicle numbers (<8). Furthermore, to maximize the ovarian response, various superovulatory treatments were compared to induce and synchronize the follicular wave emergence. Ewes were treated with progesterone intravaginal implants, plus a PGF_2α_ analog as a control, or with the same control plus estradiol benzoate, a GnRH agonist, or both; estradiol benzoate and the GnRH agonist were investigated, and their follicular wave emergence was determined by ultrasound. Follicular waves did not emerge during the studied period in 10 females, of which one was from the control group, six were from the control plus estradiol benzoate group, and three were from the control plus estradiol benzoate and GnRH agonist group. In addition, follicular emergence was less synchronized (*p* = 0.007) when estradiol was administered (103.6 h) as compared to the control (56.6 h), the control plus GnRH agonist (52.2 h), and the control plus estradiol benzoate and GnRH agonist (80.1 h). These results show that estradiol prevented, and therefore delayed, follicular wave emergence. Hence, using progesterone intravaginal implants plus a PGF_2α_ analog seems an appropriate strategy for synchronizing follicular emergency, and there are no apparent benefits of adding an GnRH agonist or estradiol benzoate [73]. Similarly, the ovarian response could also be affected by the absence or presence of CL during FSH superovulation, and the interaction between the suppressive effects of large dominant follicles [80]. Further aspects of follicular wave dynamics and synchronization have been discussed in detail by [11,81].

### 4.2. Gonadotrophins and Superovulation

Different gonadotropins exhibit varied superovulatory responses, a detailed discussion of which can be found in the introductory section of this review, and outlined in Table 1. A number of reports have shown FSH to be superior to eCG in terms of ovulation rate and embryo viability in sheep; for example, FSH produced more embryos than eCG in Suffolk ewes (5.3 vs. 3.5) [70], native Saloia sheep (5.8 and 3.6) [82], and Awassi ewes (4.83 vs. 4.66) [68] (Table 1).Moreover, few attempts have been made at comparing the multidose FSH protocol with the eCG plus FSH single injection (Table 1) [29,35]. In sum, there are discrepancies in the outcomes regarding the use of FSH in combination with eCG, whether it produces a better superovulatory response than FSH alone, and whether this response is improved by multiple injections or a single injection of eCG.

### 4.3. Dose–Response Relationship

Irrespective of gonadotropin type, the dose rate also impacts the superovulatory outcome, and variable responses have been observed, as shown in Table 2. When different doses of pFSH (250, 500, 700, and 1000 IU) were injected in a decreasing manner, 250 IU was found to be more effective than the other doses [54]. Similarly, a low dose of 128 mg of pFSH produced comparatively greater numbers of transferrable embryos (5.1) than the higher dose of 200 mg (4.4) [83]. However, some authors did not find any significant effect of dose, with viable embryos comparable between high and moderate doses of FSH; however, extreme reduction in FSH dose has been reported to cause a significant reduction in overall ovarian response [28,29,71,84] (Table 2).

These findings indicate that extreme reduction or increment in gonadotropin levels affect the superovulatory response, either by decreasing follicular recruitment and embryonic production, or inducing excessive follicle stimulation, leading to the overproduction of hormones. These processes altering endocrine profile, and sperm and oocyte transport, thereby ultimately affecting the ovarian response [6,71]. It is also suggested that a dose according to body weight can be applied, reducing the incidence of premature luteal regression and promoting the formation of apparently healthy CL, potentially increasing the efficiency of MOET while reducing the cost.

**Table 2 vetsci-09-00117-t002:** Dose of hormones and superovulation response relationship.

Dose ^1^	Hormone	Breed	OR ^2^	ER ^3^	FR ^4^ (%)	TRR ^5^	References
500	eCG	Chios	2.6 ± 0.4 ^c^	2.2 ± 0.3 ^b^	59.1	-	[85]
750			3.9 ± 0.5 ^b^	3.4 ± 0.5 ^c^	72.4	-	
1000			5.9 ± 0.8 ^a^	2.6 ± 0.4 ^b^	68.2	-	
1500			5.0± 0.9 ^a^	1.6 ± 0.5 ^a^	64.8	-	
250	pFSH	Altamurana	8.0 ± 1.5 ^b^	6.7 ± 1.4	-	4.1 ± 0.8 ^a^	[54]
500			10.2 ± 1.6 ^ab^	6.7 ± 1.5	-	2.8 ± 0.8 ^ab^	
750			13.7 ± 1.6 ^a^	8.5 ± 1.4	-	1.3 ± 0.8 ^b^	
1000			12.6 ± 1.6 ^ab^	9.4 ± 1.4	-	2.0 ± 0.9 ^b^	
12	pFSH	Chios	5.5 ± 0.7 ^eu^	4.4 ± 0.7 ^cu^	65.4 ^w^	-	[86]
16			12.4 ± 0.9 ^fy^	8.7 ± 1.0 ^dw^	67.8 ^au^	-	
12		Fresian	3.6 ± 0.4 ^ev^	2.4 ± 0.4 ^av^	95.6 ^x^	-	
16			7.0 ± 0.6 ^fz^	4.3 ± 0.8 b^x^	90.8 ^bv^	-	
90		Xinji Fine wool		6.25 ± 2.73 b		5.12 ± 2.47 ^b^	[71]
120				10.65 ± 2.21 ^a^		8.46 ± 2.25 ^a^	
150				9.55 ± 2.47 ^a^		7.82 ± 1.88 ^a^	
80	pFSH	Katahdin hair	1.4 ± 1.1	1.0 ± 0.6 ^b^	-	1.0 ± 0.5 ^b^	[84]
120			4.5 ± 1.2	3.0 ± 0.7	-	2.0 ± 0.6	
140			4.6 ± 1.1	3.0 ± 0.6	-	2.3 ± 0.5	
128	pFSH	Dorper ewes	11.3 ± 0.3 ^b^	5.9 ± 0.8	83.6 ^b^	5.1 ± 0.7	[83]
200			16.3 ± 0.3 ^a^	5.3 ± 0.8	62.4 ^a^	4.4 ± 0.7	
140	FSH	Romanov	6.3 ± 4			6.6 ± 0.5	[87]
175			7.6 ± 6			5.6 ± 0.5	
100		Santa Inês	13.5 ± 5.7	2.63 ± 2.9 ^ab^			[88]
133			9.00 ± 3.6	1.5 ± 2.5 ^b^			
200			14.88 ± 6.9	3.9 ± 3.5 ^a^			
100	pFSH	Lacaune	2.6 ± 0.7 ^a^	1.0 ± 0.5 ^a^			[28]
200			11.6 ± 1.2 ^b^	6.9 ± 1.1 ^b^			

^1^ Dose units: mg for FSH; IU for eCG. ^2^ OR, ovulation rate; ^3^ ER, embryos recovered; ^4^ FR(%), fertilization rate; ^5^ TRR, transferrable embryos. Values with different superscripts within the column show a significant difference at *p* < 0.05.

### 4.4. Breed and Age

Donor breed and age have a significant impact on the variability of the superovulatory responses of animals [89]. For example, in a study conducted over 9000 sheep flocks, different breeds showed different ovarian responses when treated with the same hormonal protocol, and this factor accounted for approximately 30% of the variability in embryo yields [90,91]. Furthermore, a higher number of CL and transferable embryos were observed in Chios than Friesian breeds when subjected to multiple doses of pFSH (Table 1) [86].

Similarly, a significantly higher number CL and embryos (13.7 and 7.9) were observed for the Rubia del Molar breed than for Negra de Colmenar (10 and 4.3) and Manchega ewes (9.8 and 6.7) superovulated with the same eight decreasing doses of oFSH [92]. Similarly, Morada Nova ewes showed a better response than Somalis Brasileira ewes interms of ovulation rate (15.38 vs. 10.56, *p* < 0.05) and embryo yield (6.79 vs. 2.90, *p* < 0.05) [9]. Contrarily, there was no difference in ovarian response between Corriedale and Bond donor ewes when injected with a split-single dose of 180 mg FSH dissolved in 10 mg/ml 750 kDa hyaluronan [36]. These results indicate that variation in ovarian response among different breeds under the same hormonal protocol may be due to the differential kinetic behavior of the exogenous gonadotrophin, follicular status and function, or environmental influences.

Furthermore, donor age also influences superovulatory responses, with maximum embryo output occurring at approximately 6 years of age [11,93]. In addition, older ewes (24–60 months) have better embryonic recovery and transferrable embryos than younger (8–12 months) ewes [15], and embryonic survival and developmental potential are also higher in older females [89]. The high superovulatory response of adult ewes could be due to the high functional capacity of the ovaries, and having undergone a sufficient development period to obtain large physicalsize [15,94]. Additionally, in prepubertal ewes, due to diminish follicular sensitivity to various gonadotropins, multiple ovulation may be less successful as compared to that in sexually mature donor females [93]. Contrarily, one study reported that the ovarian response was comparatively higher in young ewes (1–2 years old)[21] or comparable between young and adult ewes [95], which might be due to fewer disease incidences and health problems at young ages.

### 4.5. Season and Location

Sheep are seasonally polyestrous and short-day breeders, exhibiting estrus in a defined period during the year. Hence, sexual activity is affected by photoperiod through changes in melatonin secretion during the night by the pineal gland, which relays day length to the body. Sheep breeds originating from temperate climates are seasonal breeders, and use the annual variation in daily photoperiod, while those belonging to tropical regions are sexually active throughout the year [6,96]. As reviewed by [96], breeds whose origins are located between 35° N and 35° S have the tendency to breed around the year, whereas at latitudes greater than 35°, it is typical to find ewes that are seasonally polyestrous, and whose breeding seasons are initiated by declining day length. In general, the higher the latitude, the greater the photo-dependence, and the more restricted the period of breeding activity.

There is considerable evidence demonstrating the influence of season on the ovarian output in sheep as well as other livestock. Studies conducted on Lacaune ewes in France [97], Awassi ewes in Iraq [68], and Santa Inês breedin Brazil [23] found significantly higher ovarian responses during the breeding season than the non-breeding season. Moreover, although there was no difference in ovulation and quality embryos rate, total embryos recovered were increased during the breeding as compared to the non-breeding season (Table 3) [69]. Light is considered to stimulate animal reproductive function, and changes occur in the light cycle with the alternation of seasons. As such, the reproductive activity of sheep is governed by seasonal variations. A lower ovarian response may be due to the changing of season from shorter to longer days, ultimately forcing sheep to experience an annual period of reproductive quiescence in response to increased photoperiod during the late breeding season into the non-breeding season. In sheep experiencing longer darkness periods, i.e., a shifting of the photoperiod from longer to shorter days, also initiates reproductive activity due to enhancement in the secretion of melatonin. This elevated secretion of melatonin stimulates GnRH secretion, which subsequently increases the secretion of FSH and LH from the pituitary gland, resulting in the onset of ovarian activity and the beginning of the breeding season [23].

During the non-breeding season, significantly lower ovulation rate has been observed as compared to the breeding season [98]; however, higher ovulation rates were also observed in the non-breeding season [99]. Similarly, a higher ovulation rate has been recorded in the non-breeding (May) season than in the breeding season (September) in superovulated Suffolk ewes; however, the lambing and pregnancy rate was higher in the breeding season [34] (Table 3). Regarding location, experiments conducted in Katanning, Western Australia [57], United Kingdom [100], Spain [40], Portugal [12,82], and China [15] failed to detect any seasonal influence on ovarian response following superovulatory protocols. The adoption of MOET techniques, and their possible outcomes throughout the year, requires more trials to evaluate the seasonal effects on viable embryo production among different breeds.

### 4.6. Nutritional Status

The relationship between reproductive function and nutrition has not been established due to highly inconsistent ovarian responses [24,25,101,102]. Indeed, nutrition plans have shown contradictory outcomes on ovarian response. Ewes allocated 2.2 times their daily energy requirement exhibited lower ovulation rates and embryos recovery than ewes fed either 0.5 or 1.5 times their daily energy requirement [103]. Similarly, ewes achieved a higher ovulation rate and earlier embryonic development when fed 0.5 times their daily maintenance requirement, but reduced total number of viable embryos [24]. Conversely, a significantly lower ovulation rate for underfed ewes when compared with overfed and control ewes [104]. Similarly, a decreased number of follicles in underfed ewes as compared to control and overfed ewes treated with FSH [25]. Interestingly, there were found fewer proportions of cleaved oocytes and blastocysts after in vitro fertilization in overfed and underfed ewes compared to controls [105]. Under nutrition has been shown to delay embryo development after fertilization and increase embryo mortality during the first 2 weeks [103]. Nutrition changes in the hypothalamus–pituitary–gonadal axis affect embryo quality, ultimately reducing pregnancy rate [24]. Several candidates, such as hormones, growth factors and their receptors, including plasma non-esterified fatty acids, insulin-like growth factors, insulin, and leptin, are proposed as indices of metabolic signals, and are believed to send metabolic messages to the reproductive system [24]. Conversely, overfeeding has also been reported to reduce the viability of embryos collected on day 2 after fertilization [103]. In contrast, a small number of authors reported no nutritional effect on the ovarian response in sheep [102,105,106]. Additionally, there was found to be no effect of low diet (0.7) or high (1.3) daily maintenance diets on aspirated follicles and oocyte selection for cloning [106].

These results indicate that, although diet below or above maintenance requirements may not affect the ovulation rate, it does, however, greatly influence embryo recovery and fertilization rate. However, studies addressing the causes of poor embryo survival in undernourished and overfed ewes are scarce. Moreover, the molecular biology behind these processes is poorly understood, and may involve nutritional signals acting directly on the uterus, which modifies the embryo environment and thus, its survival. Some inconsistent results are likely due to differences in experimental designs, level of diet excess or restriction, and/or nutritional plane, such as length of nutritional treatment before conception/fertilization, and diet formulation [104]. Furthermore, nutritional status can be considered a key factor influencing ART efficiency. Hence, any donor and recipient will require specific nutritional management before or during any MOET program [103,104].

## 5. Strategies for Improving Superovulation Response in Sheep

### 5.1. Inhibin Immunization along with Conventional Superovulation Protocols

Inhibin is a glycoprotein hormone of gonadal origin, synthesized by the granulose cells of ovarian follicles, which exerts a regulatory effect either by suppressing FSH hormones, or the growth and maturation follicles via the intraovarian paracrine mode, or both [107]. A reduction in endogenous inhibin secretion may facilitate FSH secretion, and thus potentiate the recruitment of more follicles to enhance ovulation rate. Immunization against inhibin is useful for inducing multiple ovulation in animals [108,109,110], and has been shown to increase ovulation rate in sheep [111,112].

A significantly higher (*p* < 0.05) ovulation rate (12.1 vs. 5.0) and ova recovery (8.1 vs. 4.8) was observed in the active inhibin-immunized group than in ewes treated with conventional superovulation [112]. Similarly, a fourfold increase (*p* < 0.01) in transferrable embryos (6.7 vs. 1.5) and the proportion of quality embryos (94.6 vs. 40.6%) in ewes was observed in the inhibin-immunized group than the control group [112]. Increased ovulation rate through inhibin-immunized peptides has also been reported. For immunized heifers, the number of total embryos and transferrable embryos (15.8 and 9.6) were almost double those for heifers treated with conventional superovulatory protocol (8.3 and 5.8) [30].

Furthermore, immunization with bacterial inhibin vaccine coupled with the estrus synchronization protocol can be used as an alternative approach for increasing ovulatory follicular size and ovulation rate [110]. Similarly, a three fold increase in oocytes by the combined administration of inhibin antiserum with eCG (108), rather than using eCG (27.7) or inhibin antiserum 36.5 alone, with similar fertilization rates [113]. There is insufficient data available on sheep inhibin immunization using simplified or conventional superovulatory protocols to properly evaluate the ovarian response in terms of embryo quality and quantity, and conception rate.

### 5.2. Melatonin Implants and Superovulation

Melatonin is a natural hormone with potent free radical scavengers, and is an antioxidant and anti-apoptotic agent synthesized in the pineal gland [114,115]. Melatonin activity works in several ways to reduce oxidative stress, particularly in ovine species [116,117]. Exogenous melatonin can be administered via subcutaneous implants, which prolong the breeding season, and thus enhance the proportion of pregnant ewes. During the ovulatory process, reactive oxygen species (ROS) produced inside the follicles cause stress that ultimately induces two cell developmental blocks, resulting in poor oocyte quality [118,119], which is an important factor for sheep fertility [119]. Melatonin concentrations are several times higher in the follicles than in the serum in human preovulatory follicles [114]. This indicates that melatonin acts directly on the ovaries [120], thereby facilitating ovarian function [117,120]. Furthermore, melatonin reduces oxidative damage to mitochondrial DNA, and prevents mitochondrial oxidative stress during ovulation [121], thus improving the quality of the oocytes.

In order to improve the production of embryos, ewes treated with exogenous melatonin during the anestrous season; however, no difference in the number or quality of the embryos was observed between melatonin-treated and non-treated groups [122]. Similarly, during any MOET program, the effect of melatonin in superovulated ewes showed no effect on the ovarian response, but reduced the number of degenerated embryos during the anestrous season in the treated group, possibly as a consequence of seasonal shifts in LH secretion and/or associated effects on follicle function [100].

Studies related to melatonin implants have shown beneficial influences by protecting sheep oocytes after superovulation under various detrimental environments. The effect of melatonin implants on ovarian cyclicity and response were investigated in aged ewes. The high-prolificacy aged Rasa Aragonesa ewes treated with melatonin showed improved embryo viability after a conventional superovulation protocol [123]. Similarly, melatonin injected 36 h after CIDR removal was followed by a significant increase in estrogen level and the number of pronuclear embryos, which ultimately increased the pregnancy rate and number of lambs [119]. In addition, melatonin subcutaneous implants before superovulation and estrus synchronization at the rate of 40 or 80 mg increased the CL number (13.4 and 15.1) compared to ewes in the control group (8.8). The recovered embryos were also found to be higher in melatonin implant groups (10.3 and 10.9) than those in the control group (6.2). Similar results were also observed for embryo transfer, pregnancy, and birth rates compared to control ewes [118]. Such results have also been reported in bovines [124] and goats [123]. This demonstrates that melatonin implants have a protective effect on oocyte quality and can promote the donor’s response to superovulation, which provides more high-quality embryos for microinjection and enhances the ability of the recipients to support transgenic embryonic development. Melatonin injection in the neck during estrus increases the level of melatonin and estradiol in the blood, which has a beneficial effect on embryo production and pregnancy rate [118,119].

### 5.3. Recombinant Hormones and Superovulation

The recent use of pituitary extracts in superovulation programs, due to their FSH activity, are generally manufactured from pig and sheep pituitaries collected from abattoirs. However, the use of such pituitaries carries both positive attributes as well as disadvantages [125]. Furthermore, the use of these animal products may also introduce a risk of disease transmission among animals. Additionally, the final cost increases due to the expensive process of hormone extraction. Further still, as discussed earlier, the conventional superovulation protocols with multiple injections are not only stressful to the animal, but are also laborious, especially in large-scale production. In order to overcome these problems, researchers have attempted to replace these conventional protocols using pituitary extracts with a recombinant FSH protocol with a long half-life [125,126].

The recombinant follicle-stimulating hormone (roFSH) is the recombinant hormone that promotes superovulation in domestic animals. Exogenous roFSH promotes the maturity of multiple ovarian follicles, ultimately improving fertility as well as pregnancy rate [127]. Recently, a study using recombinant ovine FSH (roFSH) to perform superovulation in sheep and cattle reported similar results to conventional treatments [126]. A single dose of 300 µg roFSH was investigated under field conditions, and no significant difference was observed between CL (11.9 and 13.3) or total transferrable embryos (5.4 and 6.6) in Charolais and black Angus cows, respectively. Similarly, black Angus cows were treated with a single dose of roFSH, or eight doses of pFSH, and similar ovarian responses were observed [126]. Likewise, in vivo embryo production using standard pig pituitary-derived FSH via eight injections (two injections per day) was compared against the application of the extra-long-acting single-chain bovine follicle-stimulating hormone (single injection for four days) for superovulation. The number of viable embryos were 6.32 ± 0.56 vs. 6.32 ± 0.56 (*p* < 0.05) for the standard pig pituitary-derived FSH and the extra-long-acting single-chain bovine follicle-stimulating hormone, respectively [128].

Despite the above report, there is very little information on the use of a pure recombinant bovine FSH technology that could generate high superovulatory responses and improved embryo quality compared to conventional FSH. The application of a medium-acting bovine follicle-stimulating hormone with a prolonged half-life could improve the substantially the results regarding the ovarian superovulation response and the embryo quality/yield. In conclusion, the use of bovine single-chain recombinant follicle-stimulating hormone in superovulatory treatments could be an important alternative, by reducing the number of applications and offering an improved ovarian response, together with improved embryo yield and quality and superior embryo production compared to conventional FSH superovulation protocols [128].

However, recent research demonstrated that a single dose of roFSH was equally effective at inducing superovulation in beef cows and sheep, without induced secondary effects. These results suggest that roFSH could be an alternative method for increasing the pregnancy rate in ruminants. The roFSH technology in a simplified superovulation regime can be applied to sheep and other domestic animals. Additionally, further research applying different superovulation protocols for sheep could be used to determine whether roFSH given at a different dose results in a highly acceptable ovarian response and high-quality embryo production. The single administration roFSH offers the industry a safe and highly efficacious alternative to the current pituitary-derived extracts.

## 6. Artificial Insemination (AI)

AI represents the deliberate manual insertion of semen into the uterus of an adult female during estrous by a non-natural method. This technology can offer many valuable benefits, such as the prevention and control of disease spread through the eradication of direct male-to-female contact, substantial progress in enhancing the genetic exchange rate through selective breeding, and the elimination of lethal alleles [129]. However, the main limiting factor in inseminating ewes through an insemination pipette is the cervix anatomical barrier to properly transport and deposit semen. Other factors that have limited this technology in ewes include issues with semen cryopreservation, semen type (fresh or thawed), site deposition, and time of insemination, and various climatic factors, such as temperature [10,130,131,132].

After superovulation, the donor animal can be subjected to either natural mating or AI [133]. Regardless of the type of superovulation treatment, fertilization usually fails in ewes bred naturally or artificially due to the tightly folded multiple rings of the cervix. The problem of this inefficient fertilization process can be minimized by the direct intrauterine deposition of semen following exposure by surgical or laparoscopic insemination [10,133]. Increased fertilization (91.5% vs. 44.8%; *p* < 0.05) with laparoscopic AI than natural mating has been reported [63]. Additionally, a lower fertilization rate for superovulated ewes mated naturally than with laparoscopic insemination (75% vs. 82%; *p* < 0.05) [134]. However, a higher non-fertilization rate (47.1% vs. 11.5%; *p* < 0.05) for ewes inseminated through laparoscopic AI than natural mating [133].This phenomenon of lower fertilization rate during natural and laparoscopic AI may be due to failure in the transport of sperm through the tightly folded cervix, uterus or a sperm quality, or association of lower frozen–thawed semen viability, with asynchrony between ovulation time and insemination time of the commercially available semen [135].

Further comparing laparoscopic and cervical AI, there were reported about 83.3% and 60% pregnancy rates using intrauterine laparoscopic and cervical insemination, respectively [136]. Similarly, there were also reported a higher proportion (54%) of ewes with fertilized oocytes through laparoscopic AI than with cervical (19%) insemination [137]. A greater proportion of Belclare compared to Suffolk ewes yielded fertilized oocytes after cervical AI (34% versus 10%), although there was no difference after laparoscopic AI (62% vs. 60%) between the two breeds. Laparoscopic AI was found to be more effective than transcervical and vaginal AI in terms of pregnancy rate, parturition rate, lambing, and twinning rate using frozen–thawed semen; however, the study reported no differences in the outcomes of fertility results among these three methods using fresh semen. There are problems associated with fresh semen, such as short shelf-life, small numbers of superior rams, and the crucial conditions of fresh semen transportation, which encourage farmers to use frozen semen [138]. In the laparoscopic AI, semen is deposited directly into the uterine horn, which bypasses sperm transport through the cervix, thus placing the sperm in close proximity to the site of fertilization, increasing the fertilization rate [133]. Although laparoscopic AI is a high-impact technique, its relatively high cost (i.e., trained technicians and expertise, time-consuming, labor intensive, requires specialized equipment) and associated animal welfare concern has limited its use [131,137,138].

Cervical insemination results in very low rates of fertilization [139]. However, this technique produces a higher pregnancy rate using fresh semen than frozen–thawed semen [136]. The pregnancy rate following insemination of frozen–thawed semen into the vaginal fornix is comparable to that obtained following cervical insemination, and could be more applicable for using AI during large-scale production [131]. Similarly, using fresh semen, vaginal AI could be more efficient and cost-effective. However, trans-cervical AI was found to be more efficient than the vaginal method when frozen–thawed semen was used, although its efficiency was not as high as a laparoscopic method.

It is assumed that the problem of failure in fertilization following superovulation protocols can be minimized to some extent through improvements in the synchrony between ovulation and insemination procedures, and the synchrony of ovulation could be attained by inducing an LH peak; in this regard, GnRH use has been controversial [74]. Higher fertilization rates are obtained by depositing the semen into uterine horns using laparoscopic insemination [134,140]. Similarly, GnRH administered 24 h after CIDR removal, while using insemination procedures, resulted in fertilization rates of 91.9% and 86.4%, when using fresh and frozen semen, respectively. With this approach, fertilization failure does not seem to be a serious problem in small ruminants [74]. Additionally, the major factor influencing viable embryos after mating superovulated ewes is the proportion of donor ova fertilized having more than 10 to 12 ovulations [141]. Various other factors that also play a major role in outcomes of AI include body condition and prolificacy, nutritional management prior to and after insemination, hormonal protocols for estrus synchronization, ram health and age, and semen mass motility. Other factors, such as farm conditions, environment, and practitioner expertise may also affect fertility after AI in sheep [135].

## 7. Donor Embryo Recovery and Transfer to the Recipient

In small ruminants, embryo recovery from the donor animal are usually accomplished through surgical, laparoscopic, or transcervical methods [142]. This collection occurs during the morula–blastocyst stage (6–7 days after the onset of estrus) and is conducted under a dissecting microscope for evaluation; it is then further frozen or transferred to the recipient, either by laparotomy or laparoscopy [143]. Surgical-based procedures for embryo recovery from the oviduct and uterus are commonly practiced following the method suggested by [144]. Briefly, before embryo recovery through surgical procedures, to reduce post-operative intestinal adhesions, the animal should be deprived of both food and water for 24 and 12 h, respectively. The head of the donor must be placed downwards on an inclined stretcher. The uterus and oviducts should be exposed to a mid-ventral laparotomy, and the reproductive tract is flushed with differently shaped catheters with a sterile medium, as reviewed by [10].

Record the CL (functional and regressed) and large follicles on each ovary. Subsequently, recover the embryos and assess by dividing the total collected embryos by the number of CL [145]. Evaluate and classify the embryos based on their stage of development and morphological appearance as non-viable or viable. To express the percentages of recovery and fertilization, determine the result of dividing total ova by CL, and embryos by total ova, respectively. The viability rate is calculated as the percentage of viable embryos among the total number of ova obtained. After completing the process of embryo recovery, treat the wound using antibiotics after suturing the incision. This whole process may take 30 to 34 min [34].

Laparoscopy introduced later may lead to fewer adhesions, and is capable of collecting embryos from the same animal more than seven times. However, various factors, including lower average yield from single flushing than surgical collection [10,81], relatively expensive equipment requirement, and highly trained personnel, has limited its usage [142]. Thus, embryo collection through both surgical and laparoscopic processes lead to adhesions of the reproductive tract and ovaries, which can limit the number of times a sheep can be used as a donor [10], and increases the stress produced through prolonged fasting and general anesthesia [142].

Ultrasonography is a technique established in the 1990s, and has been applied to both the ovaries and the uterus of animals as a practical tool for animal production. This technique not only detects pregnancy at earlier stages, but has also made it possible to observe the development of real-time follicular dynamics of the ovary [146]. Recently, the superovulation methods in sheep have developed from upgrading hormones to technological advancements. Despite these improvements, the lack of predictability and huge variation in the response of superovulation treatments, including embryo yield/quality, continues to be the greatest problem in any MOET program [147]. In ovine MOET programs, the evaluation of CL is generally performed by laparoscopy or laparotomy [142]; however, these techniques remain invasive and traumatic, and impair the reproductive capacity of animal.

The Doppler technique has recently been studied as a means to predict ovarian responses after superovulation treatment in domestic animals. Thus, this procedure can be applied to the animals that responded poorly to hormonal ovarian stimulation [148,149]. Additionally, the welfare of animals is a growing concern in scientific research, and livestock production practices and ultrasound examinations using “clean and ethical” techniques are an excellent laparoscopy substitute for evaluating superovulation in sheep. The effectiveness of B-mode and color Doppler sonography, and serum progesterone concentrations for investigating the ovarian response in superovulated ewes, was assessed with the FSH hormone. Six days after natural mating, the total number of ultrasonographically detectable CL and luteinized unovulated follicles were strongly and positively correlated with those detected with video laparoscopy, and circulating P4 concentrations were related directly to the number of healthy CL [149].

The color Doppler technique effectively evaluates the ovarian response in superovulated ewes, and efficiently identifies animals that do not respond to superovulation. A study was conducted in order to substitute laparoscopy with color Doppler ultrasound imaging for the evaluation of CL in ewes superovulated three times at 21-day intervals. The corpora lutea were counted by color Doppler ultrasound imaging at 6days after each superovulation, and confirmed by laparoscopy 12 h later. The mean CL number was similar for both techniques, with a significant positive correlation [150]. More research is needed to identify prematurely regressing CL using ultrasonographic technology, as they could not be distinguished from healthy CL via visual assessment of either B-mode or color Doppler ultrasonograms. This has greatly contributed to our understanding of ovarian physiology, and has helped us to develop several “pin-point” protocols for hormonal treatment. Further studies are required to apply this technique, a practical tool in a research, as well as commercial, setting to aid in predicting ovarian responses and superovulatory yields in sheep.

In sheep, embryo recovery through alternative non-surgical procedures is limited due to various factors, such as the complex anatomy of the ewe’s cervix. Furthermore, the inability to perform rectal manipulation of the track usually makes the passage of the catheter/pipette through the uterus body more difficult. The non-surgical procedure was firstly demonstrated by [151], who achieved a recovery rate of 42%, while successfully recovering embryos from 11 out of 26 ewes. Attempts were made for transcervical embryo recovery, and after ripening the cervix, by injecting hormones, such as prostaglandin and estradiol; the intravaginal application of PGF_2α_, or using estradiol and oxytocin, led to improvements in the physical penetration of the cervix that facilitated transcervical embryo recovery [142]. Some success has been achieved using cervical embryos collection; however, more work is required to make it a reliable option in sheep [63,152].

Reproductive biotechnologists are working on recovering embryos non-surgically to enable the collection of embryos from a single animal at an optimum time, without any intestinal adhesion and reproductive tract complications as shown in Table 4. A success rate of 59 and 63% success rates using 50 µg cloprostenol and 200 µg misoprostol for transcervical embryo recovery, respectively [153]. The authors found this technique satisfactory for the recovery of embryos in Santa Inês sheep. Similarly, the effect of the application of intravaginal misoprostol 5 h before cervical expansion was investigated, and concluded that embryo recovery reached 95% of cervical transposition, compared with 0% for control ewes [154]. The technique was accomplished in about 30 min with a recovery of six embryos per ewe. Furthermore, there were no negative aspects of non-surgical embryo recovery.

The collected embryos are then transferred to the uterus of the oviducts of recipients by laparotomy, or using a laparoscopic technique, which is the last step of in vivo embryo production [10,142]. There are a small number of successful studies regarding embryo transfer through laparoscopy [10]. For example, a 55% success rate of cervical transposing and embryo transfer in sheep treated with dinoprostone (PGE2) [155]. In addition, 33% of pregnancies had been recorded by day 25; however, there were no fetuses detected after 30 days (55 days of pregnancy).

**Table 4 vetsci-09-00117-t004:** Efficiency of non-surgical embryo recovery during embryo transfer.

No. of Donor Animals	Fluid Recovery (%)	Duration of Uterine Flushing (min)	NSER Performed (%)	Structures/Embryos Recovered	References
17	95.6	-	59.8	6.5	[153]
58	95.7	-	94.8	6.0	[154]
23	90.1	-	80–91	1.1	[156]
16	96.2	-	78–86	6.4–7.4	[157]
23	91–94.8	24.7–26.2	80–81.5	0.5–0.8	[158]
36	97–99	21.4–25.4	83.3–100	0.7–1.0	[159]
16	95.5–97.2	27.8–29.1	81.2–87.6	2.9–4.1	[160]
36		32.6		2.5–4.8	[161]

## 8. Successive Embryo Production through Repeated Superovulation

Embryo production from the same donor animal after a specific interval via successive treatment with gonadotropins is called “repeated superovulation”. Studies based on the repeated recovery of embryos have shown detrimental effects on ovarian function and uterine cells [162], which ultimately reduce the ovarian response (Table 5). A significant reduction has been shown during in vivo embryo production after the end of the first, and start of the second, hormonal treatment, with 50 day intervals [35] and annual intervals [15]. However, no reduction in embryo recovery was observed until the third flushing by repeatedly administering sheep with conventional FSH protocol during two breeding seasons and one non-breeding season [163]. The decrease in ovarian response after successive embryo recovery was also shown during the breeding seasons of the three consecutive years. These results imply that the limited potential of the animal for repeated embryo recovery could be due to the formation of post-operative adhesions in the reproductive tract [35,163]. Due to invasive embryo procedures, the ovarian response in goats also decreased when treated with gonadotropins [94,164]. The close interval between the two successive treatments may reduce the efficiency of repeated superovulation [15]. Another primary factor that may be ascribed to the lower recovered embryos is the high production of anti-eCG antibodies by repeatedly administering eCG hormones [35] or higher quantities of FSH antibodies [94]. These results reveal that successive embryo recovery can be improved by enhancing methods of embryo recovery as well as increasing the interval between the two successive treatments. Avoiding surgical procedures, and the improvement in non-surgical recovery techniques, can not only pave the way to reduce the stress and cost of embryo recovery, but also may enable the animal to successively repeat the procedure without any reduction in ovarian response.

## 9. Conclusions

A great deal of scientific work has been conducted recently to establish a simplified protocol with promising results. Due to the prolonged exposure of FSH, FSH multiple injections have many disadvantages; therefore, focus should be directed at simplified protocols that avoid handling and stress to the animal. Researchers have given special attention to successfully enabling simplified superovulation, with some promising results in cattle. However, at present, there is insufficient data availability on simplified superovulation to make this technology more efficient and reliable in sheep. The results of simplified superovulation can possibly be improved by choosing multiparous animals with high pregnancy experience, considering ovarian follicular development activity during the initiation of FSH treatments, and the timing of the superovulatory stimulation to synchronize the timing of treatments with follicular wave emergence. FSH in combination with eCG in a single dose has shown maximum ovarian output; however, it is suggested that the dose of hormones according to body weight impacts the formation of apparently healthy CL in superovulated ewes. FSH preparations of low LH levels could be better for improving the ovarian response. More studies are required on inhibin immunization for obtaining quality embryos, their proper dosage, and their effect on the endocrine profile. Successful approaches have shown an increase in the number of embryos recovered and transferred through non-surgical procedures in goats; therefore, studies need to follow the same procedures in sheep to improve the potential for the same animal, and avoid post-operative adhesion through surgical embryo recovery. Improved knowledge of donor fertilization, collection of oocytes, and transfer of embryos, in combination with specialized equipment, highly trained personnel, expertise, and constant attention to detail may help to conserve most of the endangered species of sheep.

## Figures and Tables

**Table 3 vetsci-09-00117-t003:** Influence of season on superovulation.

Breeding vs. Non-Breeding (B vs. NB)				Reference
Ovulation Rate	Embryo Recovery Rate	Fertilization Rate	Transferrable Embryo	
8.7 ± 5.2 ^a^ vs. 8.7 ± 6.9	6.9 ± 4.3 vs. 7.1 ± 5.7	4 ± 4.2 ^b^ vs. 4.4 ± 3.6	-	[40]
21.8 ± 2.9 ^a^ vs. 9.3 ± 1.6 ^b^	7.3 ± 2.4 ^a^ vs. 3.0 ± 1.5 ^b^	-	-	[98]
9.3 ± 1.1 vs. 8.3 ± 0.8	4.7 ± 0.8 vs. 5.0 ± 0.8	-	3.2 ± 0.6 vs. 3.7 ± 0.7	[82]
12.5 ± 1.8 vs. 12.9 ± 1.4	9.5 ± 1.5 vs. 6.8 ± 1.2	96.7 ± 2.9 vs. 94.8 ± 2.4	-	[12]
8.75 ± 0.45 ^a^ vs. 4.41 ± 0.52 ^ac^	4.83 ± 0.6 vs. 3.16 ± 0.4 ^c^	-	-	[68]
5.66 ± 0.39 ^bd^ vs. 2.41 ± 0.38 ^c^	4.66 ± 0.6 ^a^ vs. 0.83 ± 0.3 ^bd^	-	-	
13.9 ± 0.8 ^a^ vs. 11.3 ± 1.8 ^a^	6.0 ± 0.5 ^a^ vs. 3.5 ± 1.0 ^b^	-	-	[69]
-	7.5 ± 4.9 ^a^ vs. 9.4 ± 6.5 ^a^	-	5.1 ± 4.13 ^a^ vs. 7.24 ± 5.6 ^a^	[15]
10.2 ± 2.94 ^a^ vs. 16.8 ± 3.23 ^b^	6.4 ± 1.70 ^a^ vs. 11.5 ± 1.7 ^b^	-	5.9 ± 1.16 ^a^ vs. 6.2 ± 1.57 ^a^	[34]
8.6 ± 6.5 ^b^ vs. 2.9 ± 4.5 ^a^	-	-	5.12 ^a^ vs. 0.29 ^b^	[23]

Values with different superscripts within the group show a significant difference at *p* < 0.05.

**Table 5 vetsci-09-00117-t005:** Ovarian response of ewes to repeated superovulatory protocols.

Breed	Ovarian Response	1st Treatment	Second Treatment	Third Treatment	Superovulatory Protocol	References
Kivircik	Ovulation rate	8.80 ± 1.3	8.75 ± 1.3	3.81 ± 0.8	Eight pFSH injections +PGF 2α; 250 µg cloprostenol	[163]
	Embryo recovery	68.7	58.8	19.6		
	Transferrable embryo	4.55	3.4	0.5		
Merino	Ovulation rate	13.8 ± 2.2 ^a^	12.1 ± 2.2 ^a^	12.0 ± 2.2 ^a^	FSH in decline dose for 3 days twice daily	[165]
	Embryo recovery	8.3 ± 1.4 ^c^	6.3 ± 1.4 ^cd^	3.9 ± 1.4 ^d^		
Ojalada	Ovulation rate	15.9 ± 2.0 ^a^	13.2 ± 1.6 ^c^	8.8 ± 1.5 ^bd^	Six injections of 6 mL (210 IU) of pFSH	[35]
	Embryo recovery	10.7 ± 1.7 ^a^	9.0 ± 1.5 ^a^	5.0 ± 0.9 ^b^		
	Fertilization rate	86	48	51		
	Transferrable embryo	7.8 ± 1.7 ^c^	3.8 ± 1.4 ^d^	2.5 ± 0.9 ^b^		
Ojalada	Ovulation rate	14.5 ± 2.1 ^a^	10.6 ± 2.3	7.5 ± 1.5 ^b^	Single injection of 6 mL (210 IU) of pFSH	
	Embryo recovery	11.3 ± 1.8 ^ac^	6.5 ± 1.6 ^d^	4.7 ± 1.4 ^b^		
	Fertilization rate	76 ^a^	53 ^b^	71 ^a^		
	Transferrable embryo	7.7 ± 1.6 ^a^	3.1 ± 1.3 ^b^	2.9 ± 1.4 ^b^		
Poll Dorset	Ovulation rate	11.50 ± 6.9	9.00 ± 5.8	6.92 ± 4.0	Folltropin-V in decreasing dose	[15]
	Transferrable embryo	7.68	6.47	6.04		
Rasa Aragonesa	Ovulation rate	11.0 ± 0.8 ^c^	11.6 ± 0.9 ^c^	8.5 ± 1.5 d	FSH in 8 decreasing dose	[166]
	Embryo recovery	7.3 ± 0.6 ^cd^	8.0 ± 0.7 ^c^	5.4 ± 1.2 d		
	Fertilization rate	79.5	77.5	63.5		
	Transferrable embryo	5.0 ± 0.6	5.4 ± 0.7	3.5 ± 0.8		
Santa Inês	Ovulation rate	7.5 ± 4.8 ^a^	3.0 ± 5.0 ^b^	2.2 ± 3.5 ^b^	200 mg FSH in multiple dose	[167]
	Embryo recovery	5.4 ± 4.4 ^a^	1.8 ± 4.0 ^b^	1.2 ± 2.3 ^b^		
	Transferrable embryo	4.0 ± 3.5 ^a^	1.2 ± 3.0 ^b^	1.1 ± 2.1 ^b^		
Santa Inês	Ovulation rate	7.2 ± 6.1 ^a^	6.3 ± 6.8 ^a^	7.5 ± 7.3 ^a^	pFSH in six decreasing dose	[23]
	Embryo recovery	1.4 ± 2.4 ^a^	2.4 ± 3.4 ^a^	1.2 ± 2.4 ^a^		

Values with different superscripts within the same row show a significant difference at *p* < 0.05.

## Data Availability

The data presented in this study are available in the manuscript.

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
