# Peer review of "Towards Improving the Outcomes of Multiple Ovulation and Embryo Transfer in Sheep, with Particular Focus on Donor Superovulation"

_vetsci, 2022, doi:10.3390/vetsci9030117_

Round 1
Reviewer 1 Report
Whilst the review has a broad coverage of the relevant literature in the area, significant revision of written english is required. Poor sentence construction, inappropriate word choice, plural and tense errors throughout. Errors in punctuation also make readability very hard. I have included a scan that identifies specific errors in up until line 211 as a guide for the authors. Note that revision and edit is required throughout the complete document.
Order of data summarised in tables should be reviewed also. Consider alphabetical order of breeds (table 1), consistent ascending order of dose, response or embryo recovery rates (table 2). Table 3 formatting (data grouping, column order and headings) should be reviewed to facilitate better data comparison across breeding and non-breeding seasons. Again Table 4 formatting (spec column 1) made interpretation of data a challenge.

Reviewer 2 Report
The present literature review proposes to explore the theme of in vivo embryo production in sheep. To this end, several topics on the subject are addressed and often data from the literature are presented in tables. I recommend publishing the review, then I make specific comments on some points of the review.
Keywords: the words presented are present in the title. I suggest change.
Introduction: the introduction presented is too long, I suggest shortening it.
avoid using PMSG and yes eCG.
Line 37: include the possibility of carrying out natural mating
Lines 69-76: authors should also consider the zero-day protocol (Menchaca).
Lines 170: What does HAP mean?
Authors should include in the review articles on the use of Doppler ultrasound in the assessment of the response to superovulation. We know that the technique is already used to replace laparoscopy.
Authors should further explore the topic of non-surgical embryo collection. You could make a table with the main results of research in the area. There is a recent article that shows the effect of cervical dilatation treatment on the embryo (https://doi.org/10.1002/vetr.1064)
Reviewer 3 Report
This is an interesting review regarding sheep superovulation protocol´s efficacy related to ovarian response and embryo collection. In general the review is well structured and comprehensive. Sometimes it needs some moderate corrections of English. I miss a couple of important points:
-First one is related to the need of a section regarding superovulation protocols in other domestic ruminant species such as cows and goats. In this section the topic can be addressed showing the differences among ruminant species with regard to the superovulation eficacy in ovarian and embryo collection responses compared to sheep species.
-Second one is the need of a final section (before the conclusions section) with regard to the future perspectives of the state of the art (e.g. new hormonal protocols, etc...). This is important due to new superovulation protocols including recombinant hormones have been developed in this and other ruminant species. Please check in the scientific literature to include citations regarding this topic including other species such as cows and goats.
Hope this helps with a new round of the manuscript reviewing process.
Round 2
Reviewer 3 Report
The manuscript has been highly improved. Thank you very much for satisfying the requeriments of the review process. Just English language and style are fine/minor spell check required.